# Body Composition According to Spinal Cord Injury Level: A Systematic Review and Meta-Analysis

**DOI:** 10.3390/jcm10173911

**Published:** 2021-08-30

**Authors:** Peter Francis Raguindin, Alessandro Bertolo, Ramona Maria Zeh, Gion Fränkl, Oche Adam Itodo, Simona Capossela, Lia Bally, Beatrice Minder, Mirjam Brach, Inge Eriks-Hoogland, Jivko Stoyanov, Taulant Muka, Marija Glisic

**Affiliations:** 1Institute of Social and Preventive Medicine, University of Bern, Mittelstrasse 43, 3012 Bern, Switzerland; oche.itodo@paraplegie.ch (O.A.I.); taulant.muka@ispm.unibe.ch (T.M.); marija.glisic@paraplegie.ch (M.G.); 2Swiss Paraplegic Research, Guido A. Zäch Str. 1, 6207 Nottwil, Switzerland; alessandro.bertolo@paraplegie.ch (A.B.); ramona.schaniel@paraplegia.ch (R.M.Z.); gion.fraenkl@paraplegie.ch (G.F.); simona.capossela@paraplegie.ch (S.C.); mirjam.brach@paraplegie.ch (M.B.); inge.eriks@paraplegie.ch (I.E.-H.); jivko.stoyanov@paraplegie.ch (J.S.); 3Graduate School for Health Sciences, University of Bern, Mittelstrasse 43, 3012 Bern, Switzerland; 4Graduate School for Cellular and Biomedical Sciences, University of Bern, Mittelstrasse 43, 3012 Bern, Switzerland; 5Department of Diabetes, Endocrinology, Nutritional Medicine, Metabolism, Inselspital, Bern University Hospital, University of Bern, Freiburgstrasse 15, 3010 Bern, Switzerland; lia.bally@insel.ch; 6Public Health & Primary Care Library, University Library of Bern, University of Bern, Mittelstrasse 43, 3012 Bern, Switzerland; beatrice.minder@ispm.unibe.ch; 7Swiss Paraplegic Center, Guido A. Zäch Str. 1, 6207 Nottwil, Switzerland

**Keywords:** spinal cord injury, paraplegia, body composition, fat composition, obesity

## Abstract

The level of injury is linked with biochemical alterations and limitations in physical activity among individuals with spinal cord injury (SCI), which are crucial determinants of body composition. We searched five electronic databases from inception until 22 July 2021. The pooled effect estimates were computed using random-effects models, and heterogeneity was calculated using I^2^ statistics and the chi-squared test. Study quality was assessed using the Newcastle–Ottawa Scale. We pooled 40 studies comprising 4872 individuals with SCI (3991 males, 825 females, and 56 sex-unknown) in addition to chronic SCI (median injury duration 12.3 y, IQR 8.03–14.8). Individuals with tetraplegia had a higher fat percentage (weighted mean difference (WMD) 1.9%, 95% CI 0.6, 3.1) and lower lean mass (WMD −3.0 kg, 95% CI −5.9, −0.2) compared to those with paraplegia. Those with tetraplegia also had higher indicators of central adiposity (WMD, visceral adipose tissue area 0.24 dm^2^ 95% CI 0.05, 0.43 and volume 1.05 L 95% CI 0.14, 1.95), whereas body mass index was lower in individuals with tetraplegia than paraplegia (WMD −0.9 kg/mg^2^, 95% CI −1.4, −0.5). Sex, age, and injury characteristics were observed to be sources of heterogeneity. Thus, individuals with tetraplegia have higher fat composition compared to paraplegia. Anthropometric measures, such as body mass index, may be inaccurate in describing adiposity in SCI individuals.

## 1. Introduction

Body composition is an important determinant of cardiometabolic disease, comprising cardiovascular disease, metabolic syndrome, and type 2 diabetes. Higher body fat, particularly central adiposity, is linked with multiple metabolic abnormalities such as impaired glucose metabolism and dyslipidemia [1,2]. Moreover, obesity is also associated with other chronic diseases, poor mental health, lower quality of life, and overall higher risk of mortality [3]. Individuals with spinal cord injury (SCI) are specifically at higher risk for obesity. Obesity and overweight were observed at rates of 29.9% and 65.8% in SCI, and were higher compared to able-bodied individuals (ABI) [4].

Changes in body composition among individuals with SCI develop rapidly after the injury. Acute changes include skeletal muscle atrophy and intramuscular fat accumulation below injury level [5]. The loss of metabolically active skeletal muscle results in decreased basal metabolic rate and resting energy expenditure [6]. The accumulation of fat leads to impaired glucose metabolism [7] and dyslipidemia [8]. Additionally, individuals with chronic injury have a sedentary lifestyle and poor compliance with dietary recommendations [9,10]. Altogether, this creates a positive energy balance that is a major determinant for fat accumulation and obesity in chronic SCI [9].

The injury level may influence not only the biochemical body processes but also the functioning of the SCI individuals. However, the relationship between injury level and body composition remains inconsistent [11]. A retrospective study showed higher obesity rates in paraplegia than tetraplegia [4], while another study showed the opposite [12]. To date, few studies have ventured into the level of injury and its impact on body composition. Most of these studies had a small sample size [13,14,15,16], and provided conflicting findings [17,18]. Previous narrative reviews did not systematically search the literature [11,19,20,21]. In addition, due to the heterogeneity of study designs included, past reviews did not provide quantitative synthesis [19,20].

We hypothesize that the injury level is an independent risk factor for body composition in SCI. If proven, this can be used to stratify and tailor the preventive measures in this group. Thus, this paper aims to synthesize the evidence on differences in body composition in persons with SCI across various injury levels, considering other injury characteristics (i.e., completeness and duration). Furthermore, we aim to identify the gaps in the literature to provide directions for future research.

## 2. Materials and Methods

This is a systematic review and meta-analysis on body composition parameters with differentiation for the level of spinal cord injury. We conducted the review following the guidelines developed by Muka et al., and adhered to the best practices in reporting in systematic reviews based on Preferred Reporting Items for Systematic Reviews and Meta-Analyses (PRISMA) [22,23]. The study protocol can be found online (International Prospective Register of Systematic Reviews, PROSPERO ID No. CRD42020166162).

### 2.1. Data Sources and Search Strategy

A medical information specialist/librarian conducted a systematic literature search on bibliographic databases—Embase (Elsevier), MEDLINE (Ovid), Cochrane CENTRAL (Wiley), and Web of Science Core Collection—from the inception of the database until 22 July 2021. A Google Scholar search was added to identify relevant supplementary records. The search strategy combined controlled vocabulary and free text words for “spinal cord injury” and “body composition”. Specifically, we looked at studies which reported body weight, body mass index (BMI), waist circumference, total body fat mass, total body lean mass, total body fat percent, trunk fat percent, trunk fat mass, trunk lean mass, leg fat percent, leg fat mass, visceral adipose tissue (VAT), and subcutaneous adipose tissue (SAT) across different levels of SCI injuries. The full search strategy can be found in the Online Appendix.

### 2.2. Study Selection and Eligibility Criteria

Our search focused on adults (≥18 years old) with SCI. We limited our search to observational studies (cross-sectional, cohort, and case-control studies). We excluded studies which used anthropometric measures to mathematically derive lean and fat mass, and focused on gold standard measures such as DEXA, thus reducing the misclassification of the outcome. We excluded studies that did not provide body composition parameters according to injury level or those which only provided beta coefficients (and other measures of association). We excluded interventional studies, animal studies, and in-vitro studies. We also excluded reviews, commentaries, letters to the editor, and conference abstracts. No language restrictions were applied.

Two authors (PFR and AB/RMZ/GF/SC/JS) screened the titles and abstracts using the defined inclusion criteria. Full-text articles were extracted and reviewed independently by two authors. In case of disagreements on study inclusion, a third author (TM/MG) was contacted if consensus between the two authors could not be achieved.

### 2.3. Data Extraction

We used a predesigned form for the data collection on the included studies. For each included study, we collected the first author’s name, setting and location, number of participants, sex, age, injury characteristics, baseline cardiovascular disease (CVD), medication use, study outcomes, and method for measuring body composition. To ensure the accuracy of the data collection, at least two authors independently extracted the data (PFR and AB/RMZ). We followed the Systeme Internationale (SI) units in the reporting of units. In studies that reported median and ranges (interquartile range, maximum, and minimum values), we converted the measures into mean and standard deviation using a previously established method [24]. Briefly, this method improved upon the estimation of means and standard deviations using medians and interquartile ranges by including the sample size in the equation. We used the author names and study location to identify double publications or analyses using similar cohorts. We also scrutinized the study methodology to determine prior reports from the same population. In these instances, the most recent publication, or the one with the most relevant outcome, was extracted (von Elm pattern of reporting) [25].

### 2.4. Risk of Bias Assessment

We used the Newcastle–Ottawa Scale for cross-sectional and cohort studies to assess the risk of bias [26]. The scale was developed to assess the quality and risk of bias in non-randomized studies. The scale uses three categories: (a) study selection (i.e., sample size, representativeness of the sample, and ascertainment of exposure); (b) comparability (i.e., factors that were compared between the groups other than the outcome); and (c) outcome (assessment and the statistical test used) [26]. The scale is from 1 to 10 stars, with a higher number of stars having higher quality. According to the scale, we classified 8–10 stars as high-quality, 5–7 stars as moderate quality, and 1–4 stars as low quality. Two authors (PFR and AB/RMZ) performed the study quality assessment. A third author (MG/TM) was consulted if the two authors did not agree with the given score.

### 2.5. Data Synthesis and Analysis

We computed pooled means and standard deviations based on the values obtained from each study (Equations (1) and (2)). We also calculated the weighted mean difference (WMD) using the DerSimonian and Laird method [27] to compare groups. In brief, this method pools the study-level estimate through the random effects model by assuming that different studies have different true effect estimates. A positive WMD signifies that the pooled mean of individuals with tetraplegia (or high paraplegia) is higher than individuals with paraplegia (or low paraplegia). A negative value means the that individuals with tetraplegia have lower pooled means compared to those with paraplegia.
(1)$$\bar{x}=\frac{\sum_{i=1}^{N}w_{i}x_{i}}{\sum_{i=1}^{N}w_{i}}$$
(2)$$\sigma =\sqrt{\frac{\sum_{i=1}^{N}{\left(x_{i}-\bar{x}\right)}^{2}}{\frac{\left(M-1\right)}{M}\;\sum_{i=1}^{N}w_{i}}}$$
where, N=number of observationM=number of nonzero weights$w_{i}=\mathrm{weights}$$x_{i}=\mathrm{observations}$$\bar{x}=\mathrm{weighted}\;\mathrm{means}$σ=weighted standard deviation


Equations (1) and (2). Weighted means and weighted standard deviation.

Heterogeneity was assessed using the Cochrane X^2^ test and the I^2^ statistic. We classified the level of heterogeneity into low (0–40%), moderate (I^2^ 30–60%) and substantial (I^2^ 50–90%) heterogeneity according to the recommendation by the Cochrane Collaboration [28]. We used stratified analyses and random-effects meta-regression, if eight or more studies were included in the meta-analysis, using the study characteristics as strata, namely, sex, age, baseline cardiovascular disease and cardiovascular medication use, duration of injury, completeness of injury, study location, number of participants, and the primary outcome of interest (whether body composition or other cardiovascular outcomes) [29]. In addition, we performed meta-regression to determine if there was a linear association between age, percentage of male participants, injury duration and completeness of injury, and body composition parameters. We also determined each outcome’s relationship to one another (e.g., BMI versus fat percentage) according to injury levels.

We used leave-one-out analyses to determine if one study was influencing the pooled estimate. This was conducted iteratively by removing one study at a time and comparing the new mean difference from the original result. We assessed for publication bias and small study effects using funnel plots and Egger’s test. All tests were performed on two-tailed tests, using a *p*-value < 0.05 as significant. Statistical analyses were performed on STATA 15.1 using the “metan” command version 9 [30]. The studies that could not be quantitatively pooled were descriptively summarized.

## 3. Results

### 3.1. Search Results

A total of 8163 potentially relevant citations were identified in five electronic databases (Figure 1). After 2639 duplicates were removed, 5524 studies underwent abstract and title screening. We assessed 331 full texts, of which we identified 57 studies reporting on body composition according to the level of injury. Forty studies (40/57, 70.2%) [10,13,14,15,16,31,32,33,34,35,36,37,38,39,40,41,42,43,44,45,46,47,48,49,50,51,52,53,54,55,56,57,58,59,60,61,62,63,64,65] could be quantitatively pooled (38 cross-sectional and 2 longitudinal analysis), which comprised 4872 individuals with SCI (Figure 1). Table 1 summarizes the study characteristics of quantitatively pooled studies, and the details of each study included in the systematic review can be found in Appendix A.

Seventeen studies (17/57, 29.8) [10,12,17,18,42,66,67,68,69,70,71,72,73,74,75] were excluded from the analysis but were qualitatively reported (Appendix A). The reasons for exclusion were: (i) seven studies failed to provide quantitative measure among different injury levels, (ii) seven studies were deemed to be similar reports of an included study, (iii) two studies cannot be pooled due to difference in reporting, and (iv) one study did not provide measures of spread (no standard deviation).

Thirty-six studies [13,14,15,16,31,32,33,34,35,36,37,39,40,41,43,45,47,48,49,51,52,53,54,55,56,57,58,59,60,61,62,63,64,65,76,77] (36/40, 90%) analyzed individuals with tetraplegia versus paraplegia, and 12 studies (12/40, 30%) investigated individuals with high- versus low- paraplegia (T5–T6 as the level of discrimination) [38,44,45,46,47,49,50,57,58,61,65,77] (Table 1). The median injury duration was 12.3 years (interquartile range (IQR) 8.03 to 14.8 years). Nineteen studies (19/40, 47.5%) included only males, 19 studies (19/40, 47.5%) had mixed-sex participants, and one study only included females (1/40, 2.5%) (Table 1). The total population included in the analysis comprised 3991 males and 825 females (56 unknown sex from one study). The median age (across studies) was 39.1 years (IQR 36 to 42.4 years). Most of the studies were conducted in the US and Canada (23/40, 57.5%) (Table 1). The majority of the studies (33/40 or 82.5%) were assessed to be high-quality (Newcastle–Ottawa score 8–10), seven were moderate-quality, and there were no low-quality studies.

For anthropometric measurements, we found 35 studies reporting BMI (35/40, 87.5%), 22 studies on body weight (22/40, 55.0%), and 11 studies (11/40, 27.5%) on waist circumference (cm) (Table 1). We found 12 studies on body fat percentage, ten studies on lean mass, and six studies on fat mass. We found six studies on regional body composition.

### 3.2. Anthropometric Measures

Individuals with tetraplegia had lower mean BMI compared to individuals with paraplegia (weighted mean difference (WMD) −0.9 kg/m^2^, 95% confidence interval (CI) −1.4, −0.5; I^2^ 86.5%) (Table 2). This was based on 32 [13,14,15,16,31,32,33,34,35,37,40,41,43,45,46,47,48,49,51,52,53,54,55,57,58,59,60,61,62,63,65,76,77] out of 36 studies comparing tetraplegia and paraplegia, which comprised 3837 participants. We found no differences when comparing high and low paraplegia (WMD −0.2, 95% CI −1.0, 0.5; I^2^ 22.7%) (Table 2). This was based on 12 studies comparing high and low paraplegia, which comprised 747 participants. Based on pooled estimates from 20 (*n* = 1299) (20/36, 55.5%) and ten studies (*n* = 752) (10/36, 27.8%), respectively, mean body weight and waist circumference did not statistically differ between individuals with tetraplegia and paraplegia.

### 3.3. Total Body Composition

Eleven studies [13,14,15,16,35,36,39,40,43,48,76], which included 1386 participants, were included in the meta-analysis of total body composition to compare tetraplegia and paraplegia (Table 2). Whole-body composition was estimated through various methods. Three studies used bioelectrical impedance [35,43,76], seven studies used X-ray absorptiometry [10,13,15,16,36,48,64], and one study used magnetic resonance imaging [14]. Individuals with tetraplegia had a higher mean fat percentage and lower mean lean mass compared to individuals with paraplegia with WMD 1.9%, (95% CI 0.6, 3.1; I^2^ 46.7%), and WMD −3.0 kg (95% CI −5.9, −0.2; I^2^ 84.1%), respectively (Table 2). Although the pooled fat mass was higher in individuals with tetraplegia than paraplegia, the mean difference did not reach statistical significance (WMD 1.7, 95% CI −0.5, 3.9; I^2^ 48.7%).

### 3.4. Regional Body Composition

We pooled five studies [13,15,16,39,40] on trunk fat composition and three studies [13,16,39] comparing leg fat composition (Table 2). Visceral adipose tissue area and volume were higher in individuals with tetraplegia compared to those with paraplegia with WMD 0.24 dm^2^ (95% CI 0.05, 0.43, I^2^ 0.0%) and 1.05 L (95% CI 0.14, 1.95, I^2^ 0.0%), respectively. Individuals with tetraplegia had higher trunk fat mass and lower trunk lean mass than individuals with paraplegia. However, the differences did not reach statistical significance. At the lower extremities, both fat percentage and fat mass were lower for individuals with tetraplegia than paraplegia, although the latter was the only one that reached statistical significance (WMD −0.03 kg, 95% CI −0.04, −0.02; I^2^ 0.0%). Only one study reported lean mass [16] on the legs, which was revealed to be lower for individuals with paraplegia than tetraplegia.

Because of differences in the outcome reported by studies using bioelectrical impedance, we pooled only the body composition findings of studies using dual X-ray absorptiometry (Appendix A). Individuals with tetraplegia had a higher total fat mass and body fat percentage compared to paraplegia, while the latter had a higher total lean mass and lower body fat percentage. However, none of these differences reached statistical significance.

### 3.5. Heterogeneity and Subgroup Analyses

We identified substantial heterogeneity (I^2^ >50–90%) on seven outcomes (38.8%) among 18 outcomes-of-interest (Table 2). In subgroup analyses and meta-regression: (i) sex was identified as a source of heterogeneity for BMI and weight; (ii) the proportion of complete injury was a source of heterogeneity for body weight; and (iii) age and duration of injury was a source of heterogeneity for waist circumference (Appendix A). Results of restriction analysis using body composition as the primary outcome can be found in the Online Appendix (Appendix A).

Restricting the analysis to studies including individuals with complete injuries, weight was higher in paraplegia compared to tetraplegia (WMD −1.7 kg, 95% CI −2.5, −0.9), while those with mixed (complete and incomplete) injuries showed no statistical difference between the groups (Appendix A). Thus, injury completeness was determined as a source of heterogeneity. For those with longer duration of injury (above the median injury duration), individuals with tetraplegia had higher waist circumference and weight but did not reach statistical significance. A reverse trend was found in those with shorter duration of injury (i.e., tetraplegia had lower waist circumference and weight compared to tetraplegia).

Furthermore, we observed a positive linear trend between increasing age and BMI (beta 0.06, 95% CI 0.004, 0.127) and waist circumference (beta 0.33, 95% CI 0.06, 0.59), and between lesion duration and weight (beta 0.57, 95% CI 0.21, 0.92) and waist circumference (beta 0.68, 95% CI 0.18, 1.19). We also observed a negative linear trend between an increasing proportion of males and weight (beta −0.21, 95% CI −0.34, −0.07), and with proportion of complete injury and weight (beta −0.09, 95% CI −0.12, −0.06) (Appendix A).

### 3.6. Meta-Regression

We also explored BMI and its relationship with other outcomes, according to the injury level, using meta-regression (Appendix A). There was a positive association between BMI and total body fat percentage for individuals with paraplegia (beta 0.23, 95% CI 0.006, 0.45), but not in individuals with tetraplegia. A positive association was observed between BMI and waist circumference for both groups (beta 0.15, 95% CI 0.02, 0.29 for individuals with tetraplegia, and beta 0.13, 95% CI 0.010, 0.26 for paraplegia). In addition, a positive association was observed between BMI and weight for both groups (beta 0.26, 95% CI 0.21, 0.31 for individuals with tetraplegia, and beta 0.27, 95% CI 0.22, 0.32 for paraplegia).

### 3.7. Leave-One-Out Analysis and Publication Bias

In the leave-one-out analyses, most of the pooled estimates were stable, with no single study changing the pooled estimate, except for a few outcomes (Appendix A). When comparing the fat mass of individuals with tetraplegia and paraplegia, removing the study of Spungen 2003 et al. would change the result from a non-significant mean difference to a significant mean difference of 3.0 kg (95% CI 0.9, 5.2) [16]. When comparing waist circumference between tetraplegia and paraplegia, the removal of the two studies [31,59] changed the estimate from a non-statistically significant mean difference to a statistically significant difference of 2.3 cm (95% CI 0.1, 4.5) and 3.2 cm (95% CI 0.6, 5.8), respectively. We found no evidence of publication bias across different outcomes as the funnel plots were symmetrical, and Egger’s tests were not significant across all outcomes (Appendix A).

## 4. Discussion

We reported higher total body fat percentage and lower lean mass in individuals with tetraplegia as compared to those with paraplegia. Regional body composition showed a more centrally located adiposity among individuals with tetraplegia as compared to paraplegia. Although the cervical injury level showed a poorer fat composition pattern, these findings were not replicated for BMI, which was lower in individuals with tetraplegia than in those with paraplegia (Figure 2).

### 4.1. Factors Underlying Differences in Body Composition According to Injury Level

Injury level may be an independent determinant of body composition in SCI. Injury level influences the extent of muscle atrophy, which defines baseline energy expenditure [6]. Individuals with tetraplegia have a lower physical activity level [78] and lower energy expenditure than paraplegia [79,80]. However, most individuals with SCI do not change their diet to match the reduced energy requirement [9,81]. This results in a positive energy balance and the accumulation of fat, which is more pronounced in individuals with tetraplegia than paraplegia [82].

We did not observe differences in body composition between high and low paraplegia. We surmise that the two groups may have similar levels of physical activity and function. The two groups may also have similar dietary patterns, thus precluding any changes in energy balance. High paraplegia has more autonomic impairment compared to low paraplegia; however, it may be insufficient to cause significant differences in body composition.

We showed that injury completeness and duration were sources of heterogeneity in our meta-analysis. Complete injury has low to absent residual motor or sensory function below the injury level. As such, complete injury has lower resting energy requirements [80], and higher fat composition, as compared to incomplete injury [16,43]. In our analysis, studies include different proportions of complete injury that caused high heterogeneity in our analysis. Aging is a strong risk factor for fat composition and obesity, owing to the physiologic changes associated with senescence. However, injury duration and aging are two variables that are difficult to disentangle as they are closely associated with each other. Previously conducted studies showed that individuals with longer duration of injury and older age have higher fat composition [16,43]. According to the injury level, we did not see any statistically significant finding in terms of waist circumference and weight. However, when stratifying by injury duration, higher waist circumference and weight were seen in tetraplegia compared to paraplegia for those with longer injury duration.

In addition, other factors, such as muscle spasticity [83] and hormonal changes [84], which are linked with injury characteristics, may also affect changes in body composition. In particular, reduced levels of testosterone, human growth hormone and insulin growth factors may reduce the capacity for maintaining lean muscle mass and strength [82,83,85]; furthermore, spasticity and skeletal muscle paralysis were linked with intramuscular fat accumulation, total body and regional fat mass, and percentage of free fat mass [82,83,85].

Finally, non-modifiable personal factors such as age and sex are important determinants of body composition and were deemed to be sources of heterogeneity in our study. Among able-bodied individuals, males have higher total body fat mass and percentage, but males have higher visceral adipose tissue [86]. For SCI, this disparity is difficult to replicate as the injury rarely occurs in females. Furthermore, aging-related changes in body composition may be more pronounced in females, especially around menopause onset, as they have higher fat accumulation and worsening cardiometabolic risk profiles at this time [87].

### 4.2. Clinical and Scientific Implication of Our Findings

In our study, we showed that individuals with tetraplegia have lower lean mass but also have a higher fat composition as compared to those with paraplegia. This finding is compatible with the combined atrophy and fat deposition that was described as neurogenic obesity in the literature [11]. More importantly, we found higher central adiposity (visceral adipose tissue and trunk fat) in individuals with tetraplegia compared to paraplegia. Thus, injury level could be an additional risk factor for obesity in individuals with SCI that is non-modifiable. It would be reasonable to tailor the nutritional and physical activity recommendations according to injury level in order to target SCI individuals who are at higher cardiovascular risk and at risk of developing obesity [88]. Up to now, clinical guidelines have not provided injury-level-specific recommendations [89,90].

Another important issue raised by our findings was the inaccuracy of BMI in assessing the adiposity of individuals with SCI. Due to its simplicity of use, BMI is a widely used screening tool to classify individuals as overweight or obese. The increase in fat accumulation and obesity in SCI was linked to higher cardiovascular risk [13], decreased functioning [91], reduced mobility [91], increased incidence of chronic pain [92], and lower quality of life in SCI [78]. In the current study, although individuals with tetraplegia had lower BMI, the group also had a higher body fat percentage, while, in our meta-regression, BMI was positively correlated with the fat percentage in individuals with paraplegia, but not in those with tetraplegia. The inaccuracy and inconsistency of BMI in estimating adiposity seen in our study are in line with previous attempts to compare the anthropometric indices for approximating body composition in SCI [20]. In the absence of a convenient, easy-to-use, and acceptable screening measure, BMI can be more accurate for individuals with paraplegia than tetraplegia.

Body adiposity is measured in research setting through bioelectrical impedance, X-ray absorptiometry, and magnetic resonance imaging, among others. Anthropometric measures and indices are more widely used in clinical settings to approximate adiposity. In populations who show similar obesity phenotypes to SCI (elderly or individuals with cancer), skinfold thickness was identified as valid proxy of body composition [93]. However, in individuals with paraplegia, the skinfold thickness may underestimate the percentage of fat in both athletic and inactive individuals with SCI [94,95]. It also fails to measure central obesity and visceral adiposity, which is more associated with cardiovascular diseases. Thus, the evidence favoring its use is still unclear. Waist circumference was positively correlated with fat percentage [96], central obesity, and visceral adipose tissue [97]. This was validated for use in SCI group [71]. However, thoracoabdominal muscle paralysis and muscle atrophy could also lead to underestimation of trunk fat mass and visceral adiposity [98]. Other indices such as the body roundness index and body shape index may be better predictors of body morphology. These indices use a combination of different anthropometric measures (e.g., height, weight, waist circumference, and abdominal circumference) and consider the different regional body compositions (trunks vs. legs) between able-bodied individuals and those with SCI. However, these indices are yet to be validated for the SCI population and need to be correlated with obesity-related diseases [96,99].

### 4.3. Strengths and Weaknesses

To the best of our knowledge, we provided the first review that compares the body composition of SCI at different injury levels. Most of the reviews on SCI were narrative and expert reviews, had no systematic quality assessment of the study, and did not synthesize the results for a pooled effect estimate [19,20]. Most SCI research has a smaller sample size due to the inherent rarity of the disease and the lack of SCI-specialized centers in most countries. Our review pooled studies comprising 4872 individuals with SCI, with 1618 having available information on fat and lean mass composition, thus providing higher statistical power and precision. We excluded studies that used anthropometric measures to mathematically derive lean and fat composition. This reduced the bias by excluding data that are highly correlated with one another. We also included studies with the level of injury and body composition as a secondary outcome to increase the number of studies and individuals analyzed, although this introduced heterogeneity into the analysis.

However, our study has some weaknesses to be considered and taken into context when interpreting our results. First, our analysis included mostly cross-sectional studies, and although most studies were evaluated as high-quality, we cannot draw any conclusions regarding causality. Second, critical factors that affect body composition (e.g., diet, sleep, mental health, physical activity) could not be integrated into our analysis as most studies did not provide standardized reporting of these factors. Our pooled estimates were based on crude measurements that were not corrected according to confounding variables. Third, body composition was measured through various methods. The pooling of the estimates resulted in significant heterogeneity in our results. Finally, the generalizability could be questioned, as most of the studies were conducted in North America, with different dietary habits and lifestyle patterns compared to other areas of the world. All studies were conducted in individuals with chronic SCI, and a large percentage of the aggregate population was also male. Thus, the estimates may not be generalizable to females with SCI or individuals in the subacute phase of the injury.

### 4.4. Knowledge Gaps and Future Research

Although current evidence indicates a more disadvantageous body composition profile in individuals with tetraplegia than individuals with paraplegia, the evidence is still of limited quality. Thus, future studies are needed to validate our findings. Specifically, when exploring the link between injury level and intermediate CVD risk factors, the completeness of the injury and injury duration and, at least, factors such as age and gender should be accounted for in the analysis (e.g., adjustment or stratification). Furthermore, all studies were conducted in chronic SCI. More longitudinal studies are still needed to investigate the trajectories of body composition changes shortly after injury (within a year following injury), and long-term follow-up is required to analyze the causal association between body fat and prevalence of cardiovascular and metabolic diseases in persons with SCI. A study on subacute SCI would provide us with a possibility of early identification of individuals with an increased risk of developing obesity, thus making it possible to promptly implement preventive strategies.

Most of the studies were conducted in North America, and were mostly based on army veterans, which does not reflect the lifestyle, diet, socioeconomic status, health systems, and various other external factors of the layperson in other regions of the world. In addition, race/ethnicity that could act as a proxy of cultural background and health behavior and could modify the outcome. We did not analyze race/ethnicity as few reported this in the baseline characteristics, thus the role of race/ethnicity could be of interest in future studies. Lastly, more female-specific studies on body composition are also needed. The current literature is predominantly based on males, albeit sex is a significant determinant of adiposity in the general population [100]. There are established hormonal changes across different reproductive stages that influence adiposity in females, and, as such, this field has yet to be explored [101].

With the rising trend in the obesity incidence in the general population, further research should focus on optimizing anthropometric screening methods (e.g., BMI, waist–hip ratio, and fat-fold analysis) for SCI. Other screening methods for obesity through non-invasive diagnostics should also be explored. Furthermore, longitudinal studies should be conducted to validate body indices for cardiovascular disease, diabetes, and other obesity-related diseases. The current cutoffs for adiposity and anthropometric measures were validated by approximating the cardiovascular disease risk in the able-bodied population [102,103], but not for individuals with SCI. These cutoffs still need to be validated before use for cardiovascular risk stratification in SCI.

## 5. Conclusions

Screening for adiposity is a priority for health promotion in individuals with SCI considering their higher risk for obesity and related complications. In this group, we observed body composition changes that are dependent on the level of injury, such that individuals with cervical levels of injury had higher total body fat percentages and visceral adiposity. Preventive measures and early intervention should be tailored according to the injury level. Furthermore, we found that BMI and waist circumference, both widely used in screening for obesity, poorly approximate adiposity. Thus, we highlighted the importance of optimizing the cutoff for anthropometric measures and emphasized the need to validate novel indices for obesity screening and prevention.

## Figures and Tables

**Figure 1 jcm-10-03911-f001:**
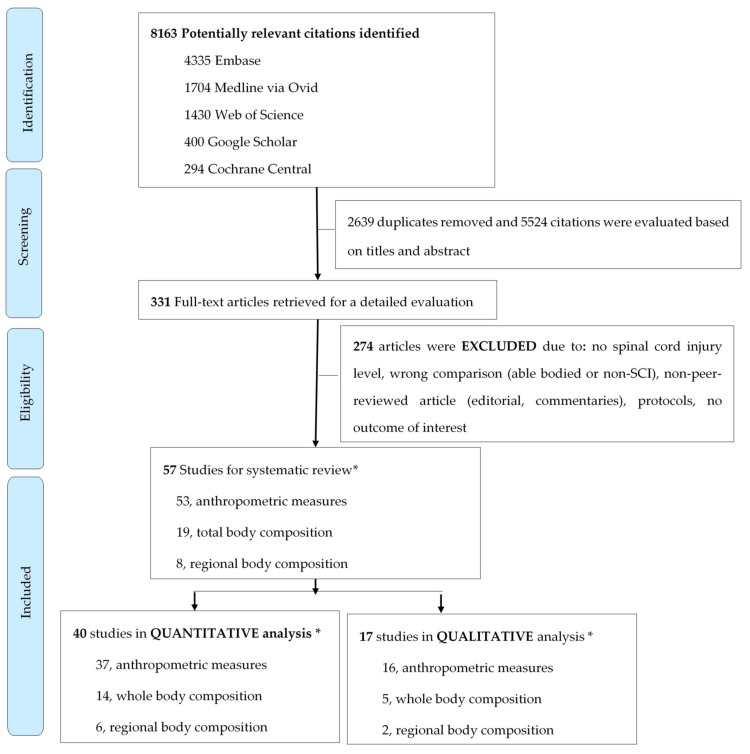
Flowchart of study inclusion (SCI: spinal cord injury; * Some studies reported information on multiple outcomes. Thus, the number of studies per outcome exceeds the total number of available studies).

**Figure 2 jcm-10-03911-f002:**
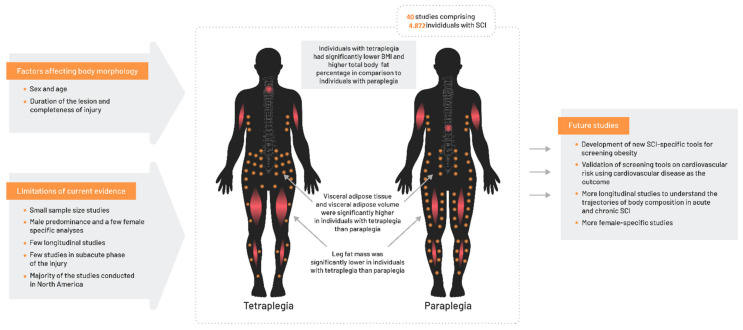
Summary of important findings.

**Table 1 jcm-10-03911-t001:** Characteristics of studies included in the meta-analysis.

Characteristics	No. of Studies	References
Level of injury ^1^		
Tetraplegia vs. Paraplegia	36	Akbal 2013 [31], Azevedo 2016 [32], Bauman 1992 [34], Bauman 1999 [33], Buchholz 2009 [35], de Groot 2008 [37], Chun 2017 [36], Farkas 2019 [40], Flueck 2020 [39], Gater 2021 [41], Gibson 2008 [76], Gomes Costa 2019 [77], Gorgey 2011A [13], Han 2015 [43], Janssen 1997 [45], Katzelnick 2019 [47], Kemp 2000 [48], Kim 2016 [49], Li 2019 [14], Matos Souza 2010 [51], McCaulay 2018 [52], Miyatani 2014 [53], O Brien 2017 [54], Pelletier 2016 [15], Powell 2017 [55], Rankin 2017 [56], Raymond 2010 [57], Ribeiro Neto 2013 [58], Sabour 2013 [59], Spungen 2003 [16], Sumrell 2018 [60], Wang 2005 [62], Wang 2007 [61], Yahiro 2019 [63], Yilmaz 2007 [64], Zhu 2013 [65]
High Paraplegia vs. Low Paraplegia	12	Dionyssiotis 2008 [38], Gomes Costa 2019 [77], Inukai 2006[44], Janssen 1997 [45], Katzelnick 2017 [46], Katzelnick 2019 [47], Kim 2016 [49], Lee 2017 [50], Raymond 2010 [57], Ribeiro Neto 2013 [58], Wang 2007 [61], Zhu 2013 [65]
Proportion of Complete Injury ^2^		
100%	12	Bauman 1999 [33], Chun 2017 [36], Dionyssiotis 2008 [38], Farkas 2019 [40], Gater 2021 [41], Gomes Costa 2019 [77], Gorgey 2011A [13], Rankin 2017 [56], Sumrell 2018 [60], Wang 2005 [62], Wang 2007 [61], Yilmaz 2007 [64]
Mixed	15	Akbal 2013 [31], Azevedo 2016 [32], de Groot 2008 [37], Gibson 2008 [76], Janssen 1997 [45], Katzelnick 2017 [46], Katzelnick 2019 [47], Kemp 2000 [48], O Brien 2017 [54], Pelletier 2016 [15], Powell 2017 [55], Raymond 2010 [57], Ribeiro Neto 2013 [58], Sabour 2013 [59], Spungen 2003 [16]
Duration of Injury (Years) ^3^		
≤12.3 y (IQR 8.03–14.8)	19	Akbal 2013 [31], Azevedo 2016 [32], Bauman 1999 [33], Chun 2017 [36], Dionyssiotis 2008 [38], Gomes Costa 2019 [77], Han 2015 [43], Katzelnick 2017 [46], Lee 2017 [50], Matos Souza 2010 [51], McCaulay 2018 [52], O Brien 2017 [54], Rankin 2017 [56], Raymond 2010 [57], Ribeiro Neto 2013 [58], Sabour 2013 [59], Sumrell 2018 [60], Wang 2005 [62], Wang 2007 [61], Yilmaz 2007 [64]
>12.3 y	18	Bauman 1992 [34], Buchholz 2009 [35], de Groot 2008 [37], Farkas 2019 [40], Flueck 2020 [39], Gater 2021 [41], Gibson 2008 [76], Inukai 2006 [44], Janssen 1997 [45], Katzelnick 2019 [47], Kemp 2000 [48], Li 2019 [14], Miyatani 2014 [53], Pelletier 2016 [15], Spungen 2003 [16], Yahiro 2019 [63], Zhu 2013 [65]
Sex		
Males Only	19	Azevedo 2016 [32], Bauman 1992 [34], Bauman 1999 [33], Dionyssiotis 2008 [38], Gomes Costa 2019 [77], Gorgey 2011A [13], Inukai 2006 [44], Janssen 1997 [45], Matos Souza 2010 [51], McCaulay 2018 [52], O Brien 2017 [54], Rankin 2017 [56], Ribeiro Neto 2013 [58], Spungen 2003 [16], Sumrell 2018 [60], Wang 2005 [62], Wang 2007 [61], Yahiro 2019 [63], Yilmaz 2007 [64]
Both	19	Buchholz 2009 [35], de Groot 2008 [37], Chun 2017 [36], Farkas 2019 [40], Flueck 2020 [39], Gater 2021 [41], Gibson 2008, Han 2015 [43], Katzelnick 2017 [46], Katzelnick 2019 [47], Kemp 2000 [48], Kim 2016 [49], Lee 2017 [50], Miyatani 2014 [53], Pelletier 2016 [15], Powell 2017 [55], Raymond 2010 [57], Sabour 2013 [59], Zhu 2013 [65]
Female Only	1	Li 2019 [14]
Unknown	1	Akbal 2013 [31]
Study Size		
<50	18	Azevedo 2016 [32], Dionyssiotis 2008 [38], Farkas 2019 [40], Gomes Costa 2019 [77], Gorgey 2011A [13], Inukai 2006 [44], Janssen 1997 [45], Katzelnick 2019 [47], Kim 2016 [49], Lee 2017 [50], Li 2019 [14], Matos Souza 2010 [51], McCaulay 2018 [52], O Brien 2017 [54], Rankin 2017 [56], Raymond 2010 [57], Sumrell 2018 [60], Yilmaz 2007 [64]
50–100	9	Akbal 2013 [31], Buchholz 2009 [35], Chun 2017 [36], Flueck 2020 [39], Gater 2021 [41], Gibson 2008 [76], Miyatani 2014 [53], Ribeiro Neto 2013 [58], Wang 2007 [61]
>100	13	Bauman 1992 [34], Bauman 1999 [33], de Groot 2008 [37], Han 2015 [43], Katzelnick 2019 [47], Kemp 2000 [48], Pelletier 2016 [15], Powell 2017 [55], Sabour 2013 [59], Spungen 2003 [16], Wang 2005 [62], Yahiro 2019 [63], Zhu 2013 [65]
Age, years		
≤39.1 y (IQR 36–42.4)	20	Akbal 2013 [31], Azevedo 2016 [32], Bauman 1999 [33], Flueck 2020 [39], Gomes Costa 2019 [77], Gorgey 2011A [13], Inukai 2006 [44], Janssen 1997 [45], Katzelnick 2017 [46], Lee 2017 [50], Matos Souza 2010 [51], McCaulay 2018 [52], O Brien 2017 [54], Rankin 2017 [56], Raymond 2010 [57], Ribeiro Neto 2013 [58], Sabour 2013 [59], Spungen 2003 [16], Sumrell 2018 [60], Yilmaz 2007 [64]
>39.1 y	20	Bauman 1992 [34], Buchholz 2009 [35], Chun 2017 [36], de Groot 2008 [37], Dionyssiotis 2008 [38], Farkas 2019 [40], Gater 2021 [41], Gibson 2008 [76], Han 2015 [43], Katzelnick 2019 [47], Kemp 2000 [48], Kim 2016 [49], Li 2019 [14], Miyatani 2014 [53], Pelletier 2016 [15], Powell 2017 [55], Wang 2005 [62], Wang 2007 [61], Yahiro 2019 [63], Zhu 2013 [65]
Location		
Europe	4	de Groot 2008 [37], Dionyssiotis 2008 [38], Flueck 2020 [39], Janssen 1997 [45]
North America	23	Azevedo 2016 [32], Bauman 1992 [34], Bauman 1999 [33], Buchholz 2009 [35], Farkas 2019 [40], Gater 2021 [41], Gibson 2008 [76], Gorgey 2011A [13], Katzelnick 2017 [46], Katzelnick 2019 [47], Kemp 2000 [48], Kim 2016 [49], Li 2019 [14], McCaulay 2018 [52], Miyatani 2014 [53], O Brien 2017 [54], Pelletier 2016 [15], Powell 2017 [55], Rankin 2017 [56], Spungen 2003 [16], Sumrell 2018 [60], Yahiro 2019 [63], Zhu 2013 [65]
South America	3	Gomes Costa 2019 [77], Matos Souza 2010 [51], Ribeiro Neto 2013 [58]
Asia	10	Akbal 2013 [31], Chun 2017 [36], Han 2015 [43], Inukai 2006 [44], Lee 2017 [50], Raymond 2010 [57], Sabour 2013 [59], Wang 2005 [62], Wang 2007 [61], Yilmaz 2007 [64]
Outcomes		
*Anthropometric Measures*		
Body Mass Index (kg/m^2^) ^4^	35	Akbal 2013 [31], Azevedo 2016 [32], Bauman 1992 [34], Bauman 1999 [33], Buchholz 2009 [35], de Groot 2008 [37], Dionyssiotis 2008 [38], Farkas 2019 [40], Gater 2021 [41], Gibson 2008 [76], Gomes Costa 2019 [77], Gorgey 2011A [13], Han 2015 [43], Janssen 1997 [45], Katzelnick 2017 [46], Katzelnick 2019 [47], Kemp 2000 [48], Kim 2016 [49], Lee 2017 [50], Li 2019 [14], Matos Souza 2010 [51], McCaulay 2018 [52], Miyatani 2014 [53], O Brien 2017 [54], Pelletier 2016 [15], Powell 2017 [55], Raymond 2010 [57], Ribeiro Neto 2013 [58], Sabour 2013 [59], Spungen 2003 [16], Sumrell 2018 [60], Wang 2005 [62], Wang 2007 [61], Yahiro 2019 [63], Zhu 2013 [65]
Body Weight ^5^	22	Azevedo 2016 [32], Bauman 1999 [33], Dionyssiotis 2008 [38], Farkas 2019 [40], Flueck 2020 [39], Gibson 2008 [76], Gomes Costa 2019 [77], Janssen 1997 [45], Katzelnick 2017 [46], Kim 2016 [49], Li 2019 [14], McCaulay 2018 [52], Miyatani 2014 [53], O Brien 2017 [54], Pelletier 2016 [15], Raymond 2010 [57], Ribeiro Neto 2013 [58], Sumrell 2018 [60], Spungen 2003 [16], Wang 2005 [62], Wang 2007 [61], Yahiro 2019 [63]
Waist Circumference (cm)	11	Akbal 2013 [31], Buchholz 2009 [35], Gater 2021 [41], Gibson 2008 [76], Inukai 2006 [44] (only in paraplegia], McCaulay 2018 [52], Miyatani 2014 [53], Pelletier 2016 [15], Sabour 2013 [59], Sumrell 2018 [60], Yahiro 2019 [63]
*Total Body Composition*		
Body Fat Percentage (%) ^6^	12	Buchholz 2009 [35], Chun 2017 [36], Farkas 2019 [40], Flueck 2020 [39], Gibson 2008 [76], Gorgey 2011A [13], Han 2015 [43], Inukai 2006 [44] (only in paraplegia), Kemp 2000 [48], Li 2019 [14], Pelletier 2016 [15], Spungen 2003 [16]
Body Lean Mass (kg)	10	Azevedo 2016 [32], Chun 2017 [36], Farkas 2019 [40], Flueck 2020 [39], Gorgey 2011A [13], Li 2019 [14], Pelletier 2016 [15], Ribeiro Neto 2013 [58], Spungen 2003 [16], Yilmaz 2007 [64]
Body Fat Mass (kg)	6	Azevedo 2016 [32], Farkas 2019 [40], Gorgey 2011A [13], Li 2019 [14], Pelletier 2016 [15], Spungen 2003 [16]
*Regional Body Composition*		
Trunk Fat Percent (%)	5	Farkas 2019 [40], Flueck 2020 [39], Gorgey 2011A [13], Pelletier 2016 [15], Spungen 2003 [16]
Trunk Fat Mass (kg)	4	Flueck 2020 [39], Gorgey 2011A [13], Pelletier 2016 [15], Spungen 2003 [16]
Trunk Lean Mass	3	Flueck 2020 [39], Li 2019 [14], Spungen 2003 [16]
Leg Fat Percent (%)	3	Flueck 2020 [39], Gorgey 2011A [13], Spungen 2003 [16]
Leg Fat Mass (kg)	2	Gorgey 2011A [13], Spungen 2003 [16]
Visceral Adiposity		
Visceral Adipose Tissue (dm^2^)	5	Gorgey 2011 [42], Li 2019 [14], Pelletier 2016 [15], Rankin 2017 [56], Sumrell 2018 [60]
Visceral Adipose Volume (L)	2	Farkas 2019 [40], Sumrell 2018 [60]
Subcutaneous Adipose Tissue (dm^2^)	2	Gorgey 2011 [42], Rankin 2017 [56]
VAT/SAT Ratio	2	Farkas 2019 [40], Sumrell 2018 [60]
Study Quality		
Moderate (5–7)	7	Bauman 1992 [34], Flueck 2020 [39], Gomes Costa 2019 [77], Inukai 2006 [44], Rankin 2017 [56], Sumrell 2018 [60], Yilmaz 2007 [64]
Good (8–10)	33	Akbal 2013 [31], Azevedo 2016 [32], Bauman 1999 [33], Buchholz 2009 [35], Chun 2017 [36], de Groot 2008 [37], Dionyssiotis 2008 [38], Farkas 2019 [40], Gater 2021 [41], Gibson 2008 [76], Gorgey 2011A [13], Han 2015 [43], Janssen 1997 [45], Katzelnick 2017 [46], Katzelnick 2019 [47], Kemp 2000 [48], Kim 2016 [49], Lee 2017 [50], Li 2019 [14], Matos Souza 2010 [51], McCaulay 2018 [52], Miyatani 2014 [53], O Brien 2017 [54], Pelletier 2016 [15], Powell 2017 [55], Raymond 2010 [57], Ribeiro Neto 2013 [58], Sabour 2013 [59], Spungen 2003 [16], Wang 2005 [62], Wang 2007 [61], Yahiro 2019 [63], Zhu 2013 [65]
Cardiovascular Diseases and Medication Use		
No Data	20	Azevedo 2016 [32], Bauman 1992 [34], Bauman 1999 [33], Chun 2017 [36], de Groot 2008 [37], Dionyssiotis 2008 [38], Flueck 2020 [39], Gater 2021 [41], Gibson 2008 [76], Han 2015 [43], Inukai 2006 [44], Katzelnick 2017 [46], Katzelnick 2019 [47], Kemp 2000 [48], Pelletier 2016 [15], Powell 2017 [55], Spungen 2003 [16], Wang 2005 [62], Wang 2007 [61], Yilmaz 2007 [64]
Only Healthy Subjects (Excluded Those with Cardiovascular Disease or Medication Use)	16	Akbal 2013 [31], Farkas 2019 [40], Gomes Costa 2019 [77], Gorgey 2011A [13], Kim 2016 [49], Lee 2017 [50], Li 2019 [14], Matos Souza 2010 [51], McCaulay 2018 [52], Miyatani 2014 [53], O Brien 2017 [54], Rankin 2017 [56], Raymond 2010 [57], Ribeiro Neto 2013 [58], Sabour 2013 [59], Sumrell 2018 [60]
Included Those With and Without Cardiovascular Comorbidities	4	Buchholz 2009 [35], Janssen 1997 [45], Yahiro 2019 [63], Zhu 2013 [65]
Primary Outcome		
Body Composition as Primary Outcome	21	Azevedo 2016 [32], Bauman 1999 [33], Buchholz 2009 [35], Chun 2017 [36], Dionyssiotis 2008 [38], Farkas 2019 [40], Flueck 2020 [39], Gibson 2008 [76], Gorgey 2011A [13], Han 2015 [43], Inukai 2006 [44], Kemp 2000 [48], Li 2019 [14], McCaulay 2018 [52], Pelletier 2016 [15], Rankin 2017 [56], Ribeiro Neto 2013 [58], Sabour 2013 [59], Spungen 2003 [16], Sumrell 2018 [60], Yilmaz 2007 [64],
Body Composition as Secondary Outcome	19	Akbal 2013 [31], Bauman 1992 [34], de Groot 2008 [37], Gater 2021 [41], Gomes Costa 2019 [77], Janssen 1997 [45], Katzelnick 2017 [46], Katzelnick 2019 [47], Kim 2016 [49], Lee 2017 [50], Li 2019 [14], Matos Souza 2010 [51], Miyatani 2014 [53], O Brien 2017 [54], Powell 2017 [55], Raymond 2010 [57], Wang 2005 [62], Wang 2007 [61], Yahiro 2019 [63], Zhu 2013 [65]

^1^ Five studies reported injury levels as tetraplegia and paraplegia, and further classified injury levels into high paraplegia and low paraplegia [47,49,58,61,65]. ^2^ Complete injury either as sensory/motor (AIS A), motor complete (AIS A-B), or clinical labeling of complete. Thirteen studies included no report on injury completeness [14,34,35,39,43,44,49,50,51,52,53,55,65]. ^3^ Three studies included no duration of injury [13,49,55]. ^4^ Thirty-five studies reported BMI, of which 32 studies compared tetraplegia and paraplegia, and 12 studies compared high and low paraplegia. ^5^ Twenty-two studies reported weight, of which 20 compared tetraplegia and paraplegia and 8 studies compared high and low paraplegia. ^6^ Twelve studies reported body fat percentage, of which 11 compared tetraplegia and paraplegia and 2 compared high and low paraplegia.

**Table 2 jcm-10-03911-t002:** Weighted mean difference of anthropometric measures and body composition among spinal cord injury patients.

Individuals with Tetraplegia vs. Paraplegia									
Outcome (Units)	Number of Studies	Study Citations	Tetraplegia, *n*	Tetraplegia, Pooled Mean (SD)	Paraplegia, *n*	Paraplegia, Pooled Mean (SD)	Weighted Mean Difference (95% CI)	I^2^	*p* Value for Heterogeneity
*Anthropometric measurements*									
Body mass index (kg/m^2^)	32	[12,13,14,15,16,31,32,33,34,35,37,40,41,43,45,47,48,49,51,52,53,54,55,57,58,60,61,62,65,76,77]	1739	24.6 (4.9)	2098	24.9 (4.5)	**−0.9 (−1.4, −0.5)**	86.5%	<0.001
Waist circumference (cm)	10	[12,15,31,35,41,52,53,60,63,76]	347	94.5 (14.4)	405	91.5 (13.4)	1.8 (−0.2, 3.9)	57.8%	0.397
Body weight (kg)	20	[10,14,15,16,32,33,39,45,49,52,53,54,57,58,60,61,62,63,76,77]	531	75.9 (15.4)	768	73.7 (14.7)	−0.3 (−1.8, 1.2)	57.8%	0.001
*Total body composition*									
Total body fat percentage (%)	11	[10,13,14,15,16,35,36,39,43,48,76]	629	34.8 (9.1)	739	32.6 (8.9)	**1.9 (0.6, 3.1)**	46.7%	0.044
Total body fat mass	6	[10,13,14,15,16,32]	177	26.8 (8.0)	226	25.1 (8.1)	1.7 (−0.5, 3.9)	48.7%	0.083
Total lean body mass (kg)	10	[10,13,14,15,16,32,36,39,58,64]	266	46.7 (7.9)	346	48.5 (8.0)	**−3.0 (−5.9, −0.2)**	84.1%	<0.000

*Regional body composition*									
Trunk fat percentage	5	[13,15,16,39,40]	171	30.6 (6.7)	226	30.3 (7.2)	−0.32 (−2.92, 2.82)	66.6%	0.017
Trunk fat mass (kg)	4	[13,15,16,39]	158	12.8 (4.5)	197	11.5 (3.9)	0.43 (−1.06, 1.92)	60.4%	0.056
Trunk lean mass (kg)	3	[14,16,39],	93	24.3 (2.2)	116	24.5 (3.2)	−0.51 (−3.11, 2.10)	81.4%	0.005
Leg fat percentage	3	[13,16,39]	92	36.0 (4.1)	128	38.6 (9.1)	−3.77 (−12.53, 4.98)	90.3%	<0.001
Leg fat mass (kg)	2	[13,16]	73	8.2 (1.0)	92	7.9 (1.8)	**−0.03 (−0.04, −0.02)**	0.0%	0.390
*Visceral adiposity*									
Visceral adipose tissue area (dm^2^)	5	[14,15,42,56,60]	96	1.52 (0.72)	119	1.25 (0.66)	**0.24 (0.05, 0.43)**	0.0%	0.858
Visceral adipose volume (L)	2	[40,60]	20	4.4 (2.4)	49	2.7 (2.0)	**1.05 (0.14, 1.95)**	0.0%	0.532
Subcutaneous adipose tissue (dm^2^)	2	[42,56]	14	1.53 (0.91)	21	1.64 (0.67)	−0.20 (−0.76, 0.35)	5.9%	0.303
VAT/SAT ratio	2	[40,60]	20	0.82	49	0.63 (0.42)	0.18 (−0.03, 0.40)	0.0%	0.832
Comparison of individuals with high paraplegia versus low paraplegia (T5–T6 as level of discrimination)									
Outcome	Number of studies		High paraplegia, *n*	High paraplegia, mean (SD)	Low paraplegia, *n*	Low paraplegia, mean (SD)	Weighted mean difference (95% CI)	I^2^	*p* value for heterogeneity
Body mass index (kg/m^2^)	12	[38,44,45,46,47,49,50,57,58,61,65,77]	460	25.1 (5.0)	287	24.3 (4.5)	−0.2 (−1.0, 0.5)	22.7%	0.221
Body weight (kg)	8	[18,45,46,49,57,58,61,77]	135	70.9 (14.4)	144	72.4 (14.8)	−0.4 (−3.7, 2.8)	0.0%	0.965
Total body fat percentage	2	[44,58]	30	23.5 (6.8)	37	22.5 (5.7)	1.3 (−1.7, 4.3)	0.0%	0.647

Abbreviations: CI, confidence interval; SD, standard deviation; VAT/SAT, visceral adipose tissue and subcutaneous adipose tissue ratio; statistically significant differences between groups are shown in bold.

## Data Availability

Not applicable.

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
