# Peer review of "Body Composition According to Spinal Cord Injury Level: A Systematic Review and Meta-Analysis"

_jcm, 2021, doi:10.3390/jcm10173911_

Round 1

Reviewer 1 Report

The manuscript conducted a systematic review of anthropometric measure and body compositions in people with chronic spinal cord injury. The review revealed inaccuracy of a traditional measure, body mass index, to assess the body compositions in this specific population compared to other measures such as adiposity and lean mass. The study was conducted with a standardized method of reviewing studies that are thorough and appropriate for this review. There is no major concern but requiring minor changes that are raised in the comments.
Recommendations:
The authors described the importance of screening methods to identify a risk of cardiovascular diseases based on some indices such as anthropometric measures. However, the authors did not introduce how impactful body indices, that can differentiate populations with high risks of cardiovascular condition from people with spinal cord injury, can improve their quality of life and social participation. In order to show the issues, the authors can add a more detailed prevalence of cardiovascular conditions for their QOL and mortality to the introduction in the first paragraph. In addition to body compositions such as fat, there are other causes of cardiovascular diseases like autonomic dysreflexia or medications. Therefore, the authors can explain more about associated factors that were not investigated in this systematic review in the introduction and the knowledge gaps and future research section.
It was not clear why high vs. low paraplegia showed no difference of anthropometric measures although paraplegia and tetraplegia showed differences. The authors can add the observations and discussion about that regardless of more impaired autonomic function in high paraplegia compared to low paraplegia, there was no difference to the section 4.1 discussion.
Lastly, although the authors emphasized the importance of taking duration since injury as an associated factor, there was no result shown beside the association between duration vs. weight and a source of heterogeneity for weight circumference. In the result section, the authors can add non-significant findings of duration of injury. Furthermore, the authors can add more discussion why duration since injury is still an important factor for body composition and health risks in this population to the discussion section. 
Minor suggestions:
There are several citation and period inconsistencies like “mentioned(26).” Page 3 145-146, Page 11 line 373-375, line 386.
Page 3 140-142 define completeness of injury, motor or sensory?
Page 8 237-238 “lean mass was lower in individuals with paraplegia (WMD -2.4 kg 95% CI -5.5, 0.6; I2 84.1%)” must be “tetraplegia”. Please correct the description.
The authors would want to add some discussion about ethnicity that may be correlated with the body composition in the 4.3 strength and weaknesses section to consider the significance.

Reviewer 2 Report

JCM-1291259

Overall: The authors aimed to do a systematic review and meta-analysis on body composition after SCI. This is a much-needed addition to the literature but before moving forward, many aspects of the paper must be adequately addressed. The paper has great oriental to make a large contribution to the literature. However, there are many issues with the paper that must be adequately address before publication. In its current form the entire manuscript is inadequately written and poorly carried out. The sloppy mistakes make the reviewer question the integrity of the manuscript, and the quality of the searches, data extraction, and analysis.

It lacks a solid introduction acknowledging the changes in body composition after SCI. In addition, if the authors aim is to focus on body composition, they need to better highlight that in the introduction. Body composition includes anthropometrics, so why specifically separate the two? There is no need to state anthropometrics as body composition is all encompassing.

The article seems to have an identity crisis. It a systemic review/meta – analysis, so why is that not indicated in the title? The authors even mention meta-analysis in the analysis section. Thus, it is.

A major issue is that the last search was conducted almost a year ago. Since this manuscript was published several other papers have been published.

The authors also do not note time since injury in their eligibility cattier. Major differences in body composition exist in acute vs. chronic injuries. This provides large heterogeneity, along with level of injury. The authors also do not discuss AIS classifications in their search– this is very important and contributes to heterogeneity. Was this factored in their searches? Or was it completely disregarded?

The authors state cohort studies were included. Cohorts are longitudinal-based studies, which the reader is told are excluded. This doesn’t make sense.

Finally, proofread your work and please consult with English Language services. There far too many run on sentences and incorrect use of tense (past vs. present). In its current form, it is like an unmentored, or poorly mentored student drafted the paper and submitted it without oversight.

Specific comments

Title: Anthropometric indices are reflected in body composition. It is sort of redundant. See comments above. The reviewer suggests changing the title to something like “Body composition according to level of spinal cord injury: a systematic review and meta-analysis”. Titles should be informative and a snapshot.  

Abstract:

Introduction:

  • Overall, the introduction is poorly written and needs to be shortened and tightened up. The authors also do a suboptimal job discussing the physiological and anatomical changes that lead to the changes in body composition after SCI. Cite current (within 5 years) primary literature in your introduction, or minimally review papers that are at least current (under five years old) and *specifically* reference your claim.
  • Line 37-38: inhabitations is the wrong word. Please substitute or remove. It isn’t needed.
  • Line 40: spinal cord injury is not a disease. It is an injury.
  • Line 42: Insert a comma before “and obesity”. Use of oxford commas are encouraged.
  • Line 43: early risk of death. Add the “risk of” before death
  • Line 43-45: poorly written sentence, especially from lines 44-45.
  • Line 47-48: Again, poorly written sentence. In fact, the section from “body composition… processes” isn’t really needed. The authors already provided and introduction to the paragraph and no new information is presented. Rewrite.
  • Line 48: what is relative intramuscular fat?
  • Line 49: remove “showing the spinal cord’s physiological influence”
  • Line 49-51: poor sentence. Not needed. It is obvious and very, very well know neurocontrol of muscle. The authors do not have a focus of neuro-fat metabolism. So the reviewer questions if this sentence is needed.
  • Line 51-53: again, poorly written sentence. Poor compliance with healthy diet recommendations. Also, citation 8 (Lieberman et al) is out of date. There are new 2020-2025 USDA dietary guidelines. Lieberman et al uses 2010 guidelines. Please review the work by Farkas et al. on nutritional health status after SCI. This is a comprehensive meta-analysis.
  • Lines 53-57. Citation 9 is a Gorgey & Gater review of obesity. This review doesn’t primarily discuss a mismatch of energy or unchanged dietary habits. There is research from Farkas et al 2019, AJPM&R that shows a level of injury-dependent mismatch in energy expenditure and intake and its effects on body composition. The authors have in fact cited it. There is no such prospective research to submit dietary habit changes after SCI. Should be taken out.
  • Lines 59-61: run on and does not make sense. The authors are poorly connecting physical function and body composition to injury characteristics.
  • Line 61-63: Use “argument”. If the argument in the literature is weak, then maybe there isn’t a relationship as the authors suggested.
  • Line 63: the citation of Gorgey et al paper is outdated. It is 2015. Please update with recent literature.
  • Line 63-64: do the authors have support stating the few articles have studied body composition/anthropometric by level of injury? Because according to this review it seems the opposite. The review counters the authors as many studies have investigated, but few have done anthropometrics. This distinction is important.
  • Line 64: mostly? Remove that word.
  • Line 67: cite *all* those narrative reviews that did not systematic search the literature.
  • Line 67-69: The authors are clearly doing a systematic review and meta-analysis. Why isn’t this in the title? The authors are also going to find heterogeneity in their data because of injury completeness. AIS C and D have significant heterogeneity.
  • Line 60-70: there have been reviews that have focused on body composition and level of injury. However, there has been no review that has *soley* focused on the influence of level of injury on body composition. Please rephrase the sentence to reflect this correction.
  • Lines 70-72: This is not a specific hypothesis. The hypothesis should focus on level of injury with reference to paraplegia and tetraplegia. Within both paraplegia and tetraplegia there is extreme heterogeneity (especially the former) that isn’t even considering the completeness of injury.
  • Line 73: body composition encompasses body size and proportions. As written, this is redundant.

Materials and Methods

  • Body composition encompasses total fat, regional fat, lean mass, regional and total lean mass, bone mineral content and density, fat free mass, etc. The authors have been very vague about what *specific* aspects of body composition they are referring to. In addition, if they are only looking at anthropometrics, which it seems like they are not doing, the paper must stop focusing specifically on anthropometrics. The authors should tighten up what they are specifically looking at. If they are looking at all things body composition that should be detailed and the constant referencing of anthropometrics should stop. More detail is needed in this section, especially inclusion/exclusion. Provide the specific equations – in text – for all calculations (for example, weighted mean difference).
  • How was weighted mean difference calculated? Tetra minus para? Para minus tetra? Provide information on what a positive vs. negative value indicates.
  • Line 78: This is more than a systematic review; this is also a meta-analysis as the authors have a statistical analysis section. Again, body composition includes anthropometric measures. Please tighten up this aim.
  • Line 79-80: this review was conducted.
  • Line 82: What does PROSPERO stand for? Write the acronym out completely.
  • Line 85: what is a medical information specialist? A Librarian?
  • Line 88: July of 2020? Why wasn’t a more recent date included for when the search was conducted? This must be updated
  • Line 89: word choice. Controlled vocabulary? What does that mean. Change the phrase.
  • Line 89-91: Earlier the authors referred to anthropometric measures as "body mass index", "waist circumference," "bodyweight", however, here they list them at the end of the sentence.
  • Line 94: adults are classified as 18 and older. “> 18” implies 18-year-old were excluded. The authors need a greater or equal to sign “≥” not a greater sign.
  • Line 95: why did the authors exclude longitudinal based studies? They authors could include baseline data. Cohort studies are longitudinal studies.
  • Line 96: why is body mass index only now being abbreviated?
  • Line 96-97: Here the authors provided specific details and what anthropometrics and what body composition variables are evaluated. This should be brought up earlier, specifically it should be written in the search strategy. Keeping the details in the supplemental file is okay, but since the search criteria isn’t too extensive it can be written in.
  • Line 100: why were studies excluded when data was derived from regression equations? The review thinks it should be included unless strongly supported by literature
  • Line 101-102: What does “We excluded studies with no disaggregated data according to the injury level.” Mean? Do the authors mean they excluded data when it wasn’t dichotomized by level of injury? Simply your writing and state what you mean.
  • Line 103: The authors state expert reviews we excluded. Where non-experts reviews included?
  • Line 105-107: Please provide the initials of the authors conducting the abstract/title review and the third offer that settled and disagreement. Also, how many disagreements were there?
  • Line 110: A pre-determined form? Please provide this form as supplemental information to the manuscript.
  • Line 111: Author? Just one author? Or Authors? Also, please the word “study” before author and then remove the word “study” from before “setting”.
  • Line 112: why were baseline CVD and medication use extracted?
  • Line 113: estimated? Measuring maybe?
  • Line 114: list the initials of the two authors
  • Line 115-116: range is listed twice. One in and one out of the parenthesis.
  • Line 116: “and” should be placed before “range.”
  • Line 116-117: Please describe how (provide equation/method) how the median and ranges, etc. were converted to mean and SD.
  • Line 117-120: this is a very subjective way to include/exclude sources. Rather than relying on the authors to report that a publication is a secondary analysis of the data. Is there a scholarly citation to support the authors method? If not, all manuscripts should be included unless otherwise indicated by the authors themselves that a paper is a secondary analysis.
  • Line 124: remove “namely”
  • Line 125: place a coma after “sample” and before “and”
  • Line 125-126: what does comparability mean? What are you comparing? The reviewer suggests including paraeneses with an “i.e.,” and a brief explanation as done earlier.
  • Line 127-128: Is the authors’ star classification systems of high, moderate, and low quality made up? Or based on literature?
  • Line 128-129: why do the authors list out the initials here, but not previously when authors are mentioned? Also, three initials are provided and the text states two authors. This doesn’t make sense
  • Line 132: this is a meta-analysis!! By simply including a data analysis section and pooling values the authors are not simply doing a systematic review.
  • Line 133: the authors must provide the specific equations used to calculate the pool values
  • Line 133-134: study arm? This isn’t a RCT. Study group (paraplegia and tetraplegia). The grouping of this data should be specified above in the extraction section. That is to say, the authors extracted data for paraplegia and tetraplegia.
  • Line 134: the remaining portion of the sentence on line 134 does not make sense. Specifically the by weights.
  • Line 135: provide information on the DerSimonian and Laird method. I.e., “Briefly, the DerSimonian and Laird method is…”
  • Line 136-137: Do you have a citation for the heterogeneity levels? Also, why isn’t a low heterogeneity level listed?
  • Line 139-140: why was a separate pooled effect estimate provided when body composition was a primary outcome?
  • Line 141: a comma is needed before “medication use”. Also, no hyphen is needed between medication use. Medication use above is mentioned without reference to cardiovascular disease. Was medication use generally extracted or what is CVD medication?
  • Line 141: stratified analysis. Stratified by what? Level of injury? Please clarify.
  • Line 145: which study characteristics mentioned? lots of characteristics have been discussed.

RESULTS

3.1 Search results. This section is ***very*** difficult to follow. The reviewer cannot make heads-or-tails out of the numbers included or excluded and Figure 1 and Table1 only make it more confusing. This section is in complete need of a revision. The reviewer suggests consulting other systematic reviews/meta-analyses to serve as an example for the writing style.

  • Also if they main objective is to compare body composition by level of injury, why not present the studies by level of injury?
  • Line 162-163: why were only 39/56 studies quantitatively pooled? What happened to the other 17 studies? Why were the excluded? This information should be added. It would be helpful if the authors also presented the number of studies as a percent of the whole number that were included. For example, the study included 30 articles that is 100%. If 15 of those 30 studies were performed in Asia than that is 50%. As written this first paragraph of the results are hard to follow.
  • Line 164: Weren’t longitudinal studies excluded? According to Table 1 it looks like this was done. But then 37+12 = 49 studies
  • Line 164: Remove “(39)”
  • Line 165: 12 out of 39 or 12 out of 56 studies? 39+12 = 51, not 56. Are the authors stating they had 51 articles that were included? This isn’t reflected in figure 1 (which is in need of a lot of work). What happened to the other 8 studies? The numbers are not clear for the total number of studies and the breakdown of the studies.
  • Line 168: list IQR in parentheses upon first abbreviating
  • Line 169: usually chronic injury?
  • Line 169 “comprised of only males” – add “of” before “only”
  • Line 170: had – not have
  • Line 169-172. This is a run on.
  • Line 159-174: overall this paragraph is atrocious. The description of the total articles and article descriptions are hard to follow and should be better reflected in another figure showing the breakdown (this would be in addition to the flowchart).
  • Line 177-179: 12 studies, 10 studies, and 6 studies out of how many studies? This is where the percent sign would be very helpful. Again, why isn’t the data presented by level of injury?
  • Lines 178-184: again, this is very hard to follow.
  • Lines 185-189: the exclusion data must come earlier so the reader knows how many papers were excluded and the final, final number of articles that were left. The authors jump around, and it is hard to follow.

3.2 – whole and regional body composition

  • Lines 194: Delete the word “The” and start the sentence with “whole-body composition”
  • Line 195: a comma is needed before “six”
  • Line 196: “using” should be “used”
  • Line 196-200: Here the authors introduce weighted mean difference (WMD) – also known as a pooled mean difference. Why wasn’t this discussed in the statistical analysis section? The results section and the statistical analysis section should follow one another. Be specific!
  • Line 197: fat percentage. Please specifically describe this as total body fat percentage. Again, Be specific. There is also a DXA measurement called regional body fat percentage.
  • Line 199: CI was never previously defined or abbreviated. Superscript the “2” in “I2” for heterogeneity. Please address.
  • Line 200-202: What was the value for pooled fat mass (present in kg)? Where is the p-value? What is the statistical significance in reference too? Heterogeneity? Or, the difference between tetra and para?
  • Line 203-204: Did the authors examine upper limb fat? If not, why?
  • Line 205: what type of units are dm2? Should that be cm2? For area?
  • Line 208: Is this statistical significance regarding heterogeneity? Or the difference between para and tetra? This must be defined throughout the paper. It is confusing.
  • Line 209: fat mass and percentage – again call it what is as listed in the table. Leg fat percentage and leg fat mass. Lower limb fat mass was greater in tetra vs. para, while % lower limb fat was less in tetra vs para. This is not surprising given that persons with tetraplegia generally have larger lower limb mass (edema, fat, etc) compared to persons with paraplegia. Consequently, this would decrease the %lower limb fat in persons with tetraplegia because you’re dividing by a smaller denominator and artificially increase the % lower limb fat in paraplegia.
  • Line 211-212: the reviewer finds it hard to believe that only one study reported on lean mass.
  • Line 216-217: obviously, if tetra have higher body fat percentage, then persons with paraplegia would have lower body fat percentage. The authors do not need to say “…lower body fat percentage.” This is obvious. This is a binary comparison.

3.3 – screening measures of adiposity

  • Why are the authors referring to BMI, waist circumference, and body weight as screening measures? These are anthropometrics. The reviewer suggests presenting these information first because discussing body composition. The presentation of the data in Table 2 should be followed in the manuscript (i.e., anthropometrics on BMI, WC, wt, then body fat mass/%, then visceral).
  • Line 222: which 31 studies? Citations?

3.4 – subgroup analyses, heterogeneity, risk of bias

  • Section should be split up into a few sections. One on heterogeneity and article quality, the other on the regression and primary outcome on body composition.
  • Please place commas after the beta value and before 95% CI. This is inconsistent.
  • What is the justification for only examining body composition metrics in studies where body comp was the primary outcome? The reviewer questions the purpose of this, especially with the results paralleling the findings from the other analysis. TO the reviewer, this sub-analysis is distracting and the authors are better doing the original analysis with all the data given a large sample size per analysis. Table 2 should have an accompanying forest plot.
  • Line 234-235: as indicated above, it is not surprising the results were in line with the overall findings. Why is this sub-analysis needed. Strong justification must be used to keep it. Otherwise the reviewer encourages taking it out.
  • Line 235: why is there a hyphen between total and body – nowhere else was this done.
  • Line 236: “Mean body fat mass and was” – typo? Missing a word?
  • Line 237: an “and” is needed before “lean”
  • Line 238-239: The “but did not reach…. Significance” can be a new sentence. This relates to the reviewer’s earlier comment about run ons. The reviewer has not take the time to edit ever sentence and run on in the manuscript. But the authors should consult with an English Language Service.
  • Line 241: what does the 38.8% refer to? A pooled estimate of what?
  • Line 241-242: Why are the authors jumping to study quality? This references Table 1. Why wasn’t that indicated?
  • Line 244: The reviewer does not recall the subgroup analyses discussed in the methods/statistical analysis section.
  • Line 247-250: run on.
  • Line 247-253: This should be a new paragraph. Maybe even its own section. Also, a comma is needed after the beta value. In some places it is included but others the authors have neglected it.
  • Line 258: Why was BMI explored? Why not %Body fat? BMI is a poor predictor of body composition in SCI (Farkas & Gater, 2017, Neurogenic Obesity).
  • Line 259-260: fat percentage? Total bod fat percentage? Be consistent throughout the paper. The authors use various measures of body fat. Thus, when they discuss I, they should be specific to the variable they are referencing to not confuse the reader (i.e., regional vs total vs lower limb body fat). Lines 265-266 seem redundant to lines 259-260. But the values are different. Please explain?
  • Lines 258-267: hard to follow. Several run on. There is also repetitive results with different values (i.e., fat and BMI). This doesn’t make sense
  • Line 267: a hyphen is needed between meta and regression.
  • Line 273-275: speculation of why Spungen et al was different belongs in the discussion. Not results. Also, calibration is a suboptimal reason. Look at the sample of participants.
  • Line 279-281: Publication bias section could be added to the section on quality of articles.

DISCUSSION

  • The entire introduction needs a rewrite. Not only is it poorly written but it doesn’t adequately reflect/synthesize the findings of the analysis. Some of the discussion is better off in the introduction as a backdrop. Again, please consult with an English Language service.
  • The first paragraph generally states the results of the meta-analysis. Largely what is stated confirms previous literature about differences in body composition by level of injury. The authors should focus the introductory paragraph of the discussion as a confirmatory analysis, rather than stating what is already known. The BMI and WC measurements have been widely known to be poor indicators of body comp in persons with SCI.
  • Line 286: The authors overuse former vs latter. Please referrer to the groups by name, and only periodically use former and latter.
  • Line 287-290: This is a run on. But more importantly, non-significant finds are non-significant. Yes, they may be higher but without significance these finds are not different between the groups and can therefore be thought of as the same. This shouldn’t be a major focus here.
  • Line 297-304: There are 4 sentences in this paragraph. Seventy-five percent of the sentences are run on sentences. In addition, this may be better suited in the introduction because the authors do not compare their findings here.
  • Line 297: “autonomic loss”. Sympathetic and sacral parasympathetic dysfunction would be better term
  • Line 299: What are “detrimental upper extremity functions”
  • Line 300: Decreased mobility leads to decreased BMR and REE? The citied sources (Alexander et al., 1995 and Mollinger et al., 1985 are very outdated and do not support this claim – especially the former article that looks at pressure injuries).
  • Line 302-304: Persons with incomplete injuries and lower levels of injury do have higher BMR
  • Lines 305-306: a source is needed noting lower physical activity levels in persons with tetraplegia.
  • Lines 307: Is this true? Cite a specific source. This is also poorly written – a run on.
  • Line 203-309: Is this true? Please cite a source
  • Line 309-310: This doesn’t make sense. Liberman et al 2014 and Perret & Stoffel-Kurt 2011 are outdated papers. Perret & Stoffel-Kurt 2011 study was not prospective it was observation so stating persons with SCI do not change their diet after their injury is not supported by this paper. Liberman was also chronic SCI. Please find more recent dietary literature in SCI. Moreover, there is very little literature that supports healthy lifestyles/dietary patterns in persons with SCI. Data on Fruits/vegetable consumption, healthy fats, complex carbohydrates, etc. is very limited (Silveira et al, 2019; Farkas et al, 2021).
  • Line 313: completely write out tetraplegia. Please do the same for line 323.
  • Line 314: are linked not were linked.
  • Line 323-324: horrible sentence. The second half from “detrimental” to “sexes” doesn’t make sense or fit with the first half.
  • Line 325: spelling of aging.
  • Line 321-329: The references listed in this section are on-SCI. Please tie this into SCI.
  • Line 332-335: Run on. It is unclear which of your several body composition findings this is referring to. It the results and discussion were clearer and better written this would make more sense. Sarcopenic obesity is a part of neurogenic obesity (Farkas & Gater, 2018, JSCM). Muscle loss and the accumulation of fat are not the only component risks contributed to neurogenic obesity after SCI.
  • Line 335-338: very important, while poorly written.
  • Line 341: which clinical guidelines? no reference is included.
  • Line 332-349: the authors really missed the mark on the clinical and scientific importance of their findings. This discussion is not novel, in fact they have been published elsewhere in several of their citied references and others (Gater, Mahr, Cowen, Nash, Dolbow, Gorgey, Farkas, McMillan, Bauman, Spungen, Taylor, etc.).
  • Line 335: does reference 43 really cover everything listed in that sentence? For lists, the authors should provide specific citations for each item. For example, please include a citation for CVD risk, decreased functioning, mobility, increased chronic pain, etc. As written the reviewer isn’t convinced.
  • Line 356-358: This is important. BMI and Fat were not related in tetra.
  • Line 360: Laughton et al cutoff is for obesity, not overweight.
  • Line 362: medical societies/ which medical societies? Nash et al in the current CPG guidelines? The only guidelines that recommend that are PVA CPG.
  • Line 366-370: Please see Gater et al, 2021, TSCIR.
  • Line 365-373: Skin folds, along with several other listed measures do not estimate visceral adipose tissue/central obesity which is strongly associated with cardiometabolic dysfunction. Several anthropometric models do not account for restrictive lung disease (i.e., abdominal and thoracic muscle paralysis) after SCI. This impacts measurements and cutoffs of abdominal fat after SCI. The authors missed the mark on discussing the issue with anthropometrics after SCI. Also, what about the underestimation of fat in tetraplegia? It is likely worse for reasons mentioned above.
  • Line 387-379: Gill S et al, 2020 validated an SCI specific cutoff for waist circumference.
  • Sections on strengths/weakness and knowledge gaps/future research should be rewritten and revised after the paper is updated. Overall, deeper thought into the analysis and literature are required here and in the rest of the discussion. A better and updated evaluation of literature is also needed to synthesize the current finds.
  • Line 457-458: Lean mass wasn’t significant. Why do the authors keep mentioning it? Or, is the reviewer mistaken? It is hard to interpret.

Generally, figures and tables should be standalone items in a manuscript. A reader should be able to reference only the figures and tables and understand most (if not all) of the paper. In the current form this paper does not follow that guideline. The following critique highlights several weaknesses and other issues in data presentation. Please be sure the inline text of the results follows the tables and figures, and vice versa.

TABLE 1:

  • There are many issues with this table. Why aren’t the specific references for the papers listed in Table 1 also in the citation list? This *must* be done. Several authors have more than one first author publication that comes out in the same here. Thus, without being able to specifically reference that paper uncertainty remains. DO not make the reader guess.
  • Provide specific references. High vs. low should be defined in the box or in a legend.
  • Complete injuries refer to AIS A (sensory and motor complete). Authors like Farkas, Gorgey, Rankin, Bauman, Sumrell, Yilmaz include motor complete (AIS A and B). This is completely wrong. Bauman is also spelled wrong.

TABLE 2

  • Again, as indicated above, superscript the “2” in “I2”. Also please superscript other numbers where appropriate. Sloppy mistakes.
  • If the authors are comparing tetra vs. para, why aren’t those classifications used in the table? Why are instead high injury vs. low injury used?
  • The authors need to add the citations to table 2. For example, leg fat mass is discussed in 2 articles. Please the citations. The authors can simply use the same number formatting from the in-text article and add it to the table using whatever reference manager they are using.
  • A legend needs to be added to this table defining VAT, SAT, etc. Also are the units correct for VAT and SAT? Usually, it is measured in volume or cross section area. What is dm2? Is that supposed to be cm2? Where are the data for SAT volume (in L)?
  • It appears the authors are statistically comparing tetra vs. para but the only p-value is the I2. The authors should provide the p-value corresponding to the group comparisons.

FIGURE 1: There are several issues with this figure that make it suboptimal for publication. There is significant detail missing from this figure and the numbers do not add up – literally

  • The methods state embase.com (Elsevier), MEDLINE (Ovid), Cochrane CENTRAL 86 (Wiley), and Web of Science Core Collection were searched, as well as google scholar. But in the figure only embase, web of science, and Medline are provided. What happened to the other databases?
  • In its current form the total articles add up to 6712, not 7411. This is very sloppy and makes me question the integrity of the manuscript. In addition, with 7411 studies and 2584 removed, the net amount does not add up to 4863, rather it is 4827. The data discussing why articles were excluded should also be mentioned in text.
  • 56 studies were included for the systematic review? 52 for screening measures + 15 for total body comp + 8 for regional body composition do not total 56. These numbers do not make sense.

FIGURE 2

  • The order that the data is presented here should match how the text is written is section 3.4 (i.e., present screening measurements, total body composition, regional body comp, and then visceral adiposity). Otherwise change the figure. A similar comment was provided for table 2 and its corresponding text.
  • In figure 2 the authors use “no. of tetraplegic individuals” and “no. of paraplegic individuals”. But in Table 2 they use high vs. low. Why the difference. The lack of consistence as noted earlier is sloppy and makes the reviewer question the validity of the data.
  • Define VAT and SAT in a legend
  • Can an overall weighted mean be added to the figure? Why are some diamonds black and others are white? A legend would clarify this.
  • The figure capture states body composition as an outcome. Is this a main/primary outcome? Be specific!

FIGURE 3

  • Is this a graphical abstract or a figure for the paper? if it is the graphical abstract why is it referenced in the paper and why is it placed in line with the text?
  • Individuals looks weird in the upper box denoting number of articles and sample size. Speaking of article numbers, the reviewer is still uncertain how many articles were actually included in the review vs meta-analysis.
  • Lean mass was not evaluated in the upper limbs. The figure demonstrates larger upper limb lean mass in persons with Para vs Tetra. Please remove lean mass from the upper limbs in this figure. Only variables discussed in this paper should be presented in the figure.
  • The text in the center of the figure pointing to persons with tetra and para and denoting visceral fat, trunk fat, and leg fat should be separated. Trunk and visceral fat should correctly point to their respective locations, while leg fat mass should point to the legs. In addition, mention of significance should also be included in this figure. Was visceral fat, leg fat mass, etc significant greater in tetra vs. para? Do not provide the p-value, but rather include the word significant (or not significant when something isn’t). Leg fat pass is only shown in the thigh. Why not the actual leg (knee to ankle) – the authors should demonstrate that the entire leg (thigh, leg) is fatter.
  • On the right, risk is misspelled with a “u”
